# Gravity influences bevacizumab distribution in an undisturbed balanced salt solution *in vitro*

**Rae Young Kim[1], Soonil Kwon [2]\*, Ho Ra [1]\***

**1** Department of Ophthalmology & Visual Science, College of Medicine, The Catholic University of Korea, Seoul, Republic of Korea, **2** Department of Ophthalmology, Hallym University Sacred Heart Hospital, Anyang, Republic of Korea

\* raho@catholic.ac.kr (HR); magicham@hallym.or.kr (SK)

## Abstract

### Purpose

The effects of gravity on bevacizumab or the recommended head position after intraocular bevacizumab injection have not been reported. To evaluate the effect of gravity on bevacizumab *in vitro*, we added bevacizumab to the upper part of a test tube filled with balanced salt solution (BSS) and examined its distribution over time.

### Materials and methods

Sixty-four test tubes were divided equally into two groups; group 1 (32, collected from upper part of the tube) and group 2 (32, collected from lower part of the tube). Each test tube was filled with 5 mL BSS before bevacizumab (1.25 mg/0.05 mL) was added, and then stored at 36˚C. Bevacizumab concentration in 8 test tubes from each group was measured at 12, 24, 48, and 168 h using an enzyme-linked immunosorbent analysis (ELISA) kit. Mann–Whitney and Jonckheere–Terpstra tests were used for statistical analysis.

### Results

Bevacizumab concentration was significantly higher in Group 2 than in Group 1 at 12, 24, 48, and 168 h (12, 24, 48, and 168 h; $P < 0.01$ each; Mann–Whitney test). The mean change in bevacizumab concentration over time tended to increase in Group 1 ($P < 0.01$; Jonckheere–Terpstra test), but tended to decrease in Group 2 ($P < 0.01$; Jonckheere–Terpstra test).

### Conclusions

The significant differences in concentration between the upper and lower parts even after a considerable amount of storage time showed that bevacizumab did not dissolve immediately and diffused evenly throughout the solution. It appeared that more bevacizumab settled in the lower part of the tube than in the upper part because of gravitational force. However, the concentration difference between the upper and lower parts decreased as bevacizumab gradually diffused over time, indicating that the difference in concentration due to gravity was more significant at the beginning of bevacizumab injection.

**Data Availability Statement:** All relevant data are within the manuscript and its Supporting Information files.

**Funding:** The author(s) received no specific funding for this work.

**Competing interests:** The authors have declared that no competing interests exist.

## Introduction

Proliferative diabetic retinopathy (PDR) can cause vitreous hemorrhage, retinal detachment, and neovascular glaucoma, and can even lead to blindness [1]. Vitrectomy is a surgical treatment for restoring vision in PDR patients [2]. At the end of vitrectomy in PDR patients, bevacizumab (Avastin®; Genentech, San Francisco, CA, USA) is injected intraocularly as a relatively safe method to effectively reduce recurrent intraocular hemorrhage [3]. Moreover, intraocular bevacizumab injection is known to be a safe and effective method to treat vitreous hemorrhage occurring after vitrectomy for PDR patients [4]. For this reason, bevacizumab injection is often administered after vitrectomy in patients with PDR. We also typically administer an intraocular bevacizumab injection after vitrectomy in patients with PDR, and then observe using a surgical microscope whether the injected drug collects at the bottom of the eye. So, we questioned whether bevacizumab dissolves and disperses rapidly and evenly throughout the vitreous chamber when injected into balanced salt solution (BSS)-filled eyes after vitrectomy.

To date, there have been numerous studies on the half-life, clearance rate, and overall pharmacokinetics of intraocular bevacizumab injections after vitrectomy, but there have not been studies on the intraocular distribution by weight [5–7]. Hence, using an indirect method initially, we examined the distribution of bevacizumab over time following its addition into the upper part of a test tube filled with BSS. We aimed to provide insights into the intraocular distribution of bevacizumab by weight following post-vitrectomy injection, and to present our opinions about postoperative head position.

## Materials and methods

Sixty-four test tubes were divided into two groups of 32 test tubes each. Each borosilicate glass test tube (diameter: 10mm, length: 100mm) was filled with 5 mL of BSS (BSS Plus, Alcon laboratories, Fort Worth, TX). A 1-cc syringe with a 30-gauge needle was used to inject 1.25 mg/0.05 mL bevacizumab (Avastin; Genentech, San Francisco, CA, USA). By using the 1-mL syringe positioned vertically with the needle downwards from 0.5 cm above the surface of the solution in each test tube. Bevacizumab was added at a rate of two drops per second while carefully making sure that the needle did not touch the tube's internal wall. Following the drug injection, the tubes were immediately covered and stored undisturbed in $CO_2$ INCUBATOR (MCO 175, Sanyo, Japan) at a temperature of 36.0˚C and $CO_2$ of 0.0%. The concentration was measured in eight tubes from each group after 12, 24, 48, and 168 h of storage. The solution was collected only once from each tube: the upper part was collected from one tube and the lower part from another tube. For the upper layer, the solution was collected 0.5 cm below the surface, and for the lower layer, the solution was collected 0.5 cm above the floor. From the upper (Group 1) and lower (Group 2) parts of the tube, 0.2 mL of solution was collected by using a micropipette to avoid dispersion of the drug. Microcapillary tips (Denville scientific, Metuchen, NJ) were used to collect the lower layer of the solution, while carefully minimizing disturbances of the upper layer or its mixture with the lower layer. Bevacizumab's concentration was analyzed using an enzyme-linked immunosorbent analysis (ELISA) kit (Protein Detector ELISA Kit; KPL, Inc., Gaithersburg, MD, USA), which was used immediately after calibration of the concentration according to the manufacturer's calibration guidelines.

Statistical analysis was performed using IBM SPSS 21.0 software (SPSS Inc., Chicago, IL, USA). *P*-values < 0.05 were considered statistically significant.

## Results

Bevacizumab concentrations in the samples collected from the 64 test tubes [Group 1 (32, from the upper part) and Group 2 (32, from the lower part)] were analyzed. No samples were lost or contaminated. Table 1 and Fig 1 show concentration of bevacizumab over time in the upper and lower parts of the test tube following bevacizumab injection. The details of individual concentration measurements for each tube at each time point are shown in S1 Table. We compared the concentrations in Groups 1 and 2 at 12, 24, 48, and 168 h. A significant difference was observed between the two groups at all points, with a higher bevacizumab concentration in Group 2 than in Group 1 (12 h, $P < 0.01$; 24 h, $P < 0.01$; 48 h, $P < 0.01$ and 168 h $P < 0.01$ by Mann–Whitney test). The mean changes in bevacizumab concentration in Group 1 and Group 2 were calculated. The bevacizumab concentration in Group 1 showed a significant increasing trend over time ($P < 0.01$ by Jonckheere–Terpstra test), while Group 2 showed a significant decreasing trend ($P < 0.01$ by Jonckheere–Terpstra test). The difference in bevacizumab concentration between the two groups showed a gradually decreasing pattern over time, but this decrease was not statistically significant ($P = 0.275$ by Jonckheere–Terpstra test).

## Discussion

This study showed that the bevacizumab concentration in the lower part of the tube was significantly higher than that in the upper part of the tube from 12 to 168 h post injection, even when the drug was injected from above the surface.

The significant differences in concentration between the upper and lower parts even after a considerable amount of time had passed showed that bevacizumab does not dissolve immediately and diffuse evenly throughout the solution. It appears that more bevacizumab settles in the lower part of the tube relative to the upper part because of gravitational force. Furthermore, the mean concentration of bevacizumab tended to significantly increase in Group 1 over time, but tended to significantly decrease in Group 2, indicating that the concentration difference between the upper and lower parts decreased as bevacizumab gradually diffused over time. Thus, this indicates that the difference in concentration due to the effect of gravity is more significant particularly at the beginning of bevacizumab injection.

Several factors affect the change in bevacizumab concentration when it is injected directly into the vitreous chamber, including convection currents, diffusion, temperature, and volume. Jooybar et al. reported that during intravitreal injection, the position, needle size, and injection speed could cause differences in the distribution of the injected drug within the vitreous chamber [8]. In the present study, the samples were stored close to body temperature. The test tubes were filled with 5 mL of BSS, which is equivalent to the vitreous volume. Bevacizumab at a concentration of 0.05 mL, which is consistent with the amount used for intraocular injection, was

**Table 1. Mean bevacizumab concentrations in samples collected from the upper (Group 1) and lower (Group 2) parts of the test tube.**

| Group | Concentration (µg/mL) | | | |
|---|---|---|---|---|
| | 12 h | 24 h | 48 h | 7 days |
| **Group 1** | 67.28 ± 19.61 (33.5–82.4) | 86.69 ± 19.85 (50.4–110.5) | 73.04 ± 17.64 (43.4–98.5) | 103.93 ± 14.61 (74. 2–120.4) |
| **Group 2** | 443.33 ± 31.39 (403.6–426.70) | 452.44 ± 25.31 (412.4–488.8) | 442.06 ± 24.14 (405.6–466.7) | 393.90 ± 28.04 (348.3–421.5) |
| *Difference | 376.05 | 365.75 | 369.02 | 289.97 |

Values are mean ± standard deviatio (range), unless otherwise indicated

*Difference = mean value of Group 2 –mean value of Group 1

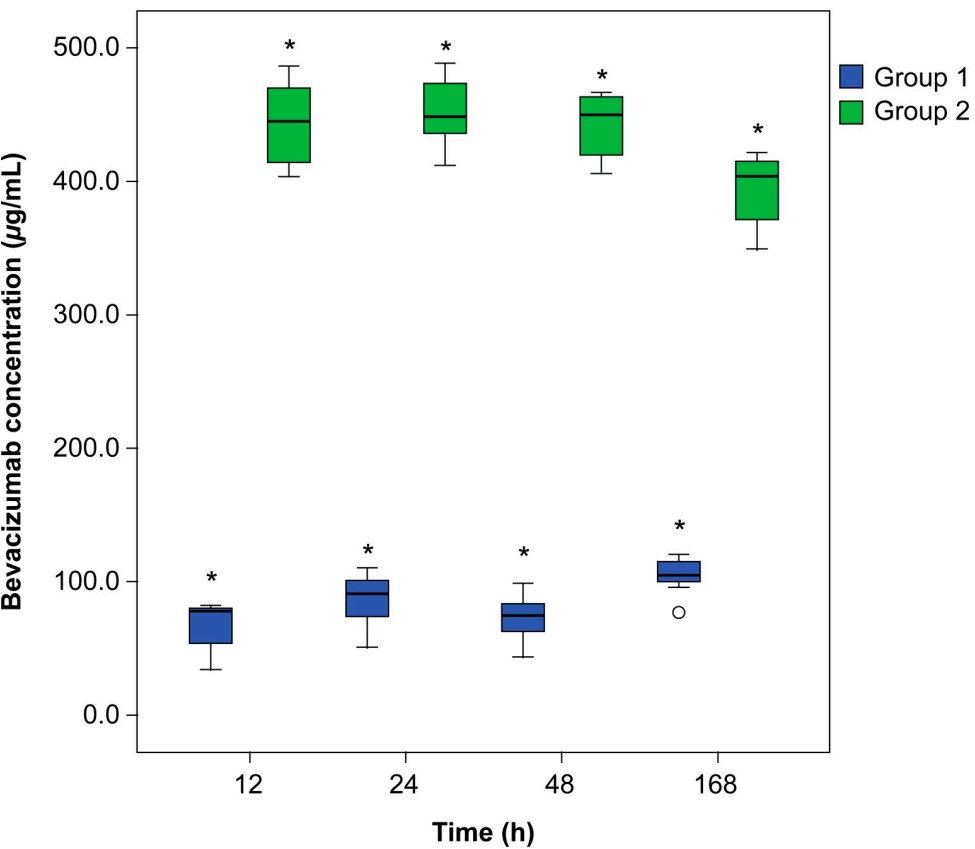

**Fig 1. Box plot of bevacizumab concentrations in Group 1 (upper part of the tube) and Group 2 (lower part of the tube).** \*: $P < 0.001$. *P* values were calculated using the Mann–Whitney test.

injected using a needle of constant size and at a constant injection speed. Under these conditions, we observed a significant difference between the upper and lower parts of the mixture over a long period, which suggests that similar results may occur in clinical practice.

Although no research on the effect of gravity on bevacizumab has been conducted to date, previous studies have reported the effects of head position and gravity on the intraocular injection of other drugs. Lim et al. administered intravitreal injections of gentamicin, which is heavier than BSS, in rabbits that underwent vitrectomy. The eyeballs were collected after keeping the rabbits in a fixed position for 30 min, and then evaluated. The study reported significant injuries to the retinal tissue located inferiorly and indicated that gravity affects intravitreal injection of gentamicin. According to these results, they recommended that patients should be placed in an appropriate position during intravitreal injection to minimize foveal damage [9]. Jaissle et al. reported that gravity could lead to the accumulation of crystals in the inferior part of the vitreous humor following intraocular triamcinolone injection. The study showed that deposition could occur at the posterior pole of the retina depending on the patient's head position, and this form of deposition could be more likely after vitrectomy [10]. Of course, while it requires the results of additional animal experiments or *in vivo* clinical trials, bevacizumab showed a higher concentration at the inferior side due to gravity as demonstrated in our study. We believe that these factors can be considered similar to the above drug during intraocular injection of bevacizumab after vitrectomy. For example, if the patient is asked to maintain a supine position after injection, a higher concentration of the drug could be delivered to the

fovea. In case of a problem in the anterior eye, such as neovascular glaucoma, the patient could be made to keep prone position after injection.

The limitation of this study is that the experimental conditions were not identical to real-world clinical situations. The surface area/length/shape of the tubes used in this study are different from that of an actual vitreous chamber. Moreover, patients are not in a completely fixed, stationary position in real-world conditions, and their posture or movement could cause the solution to mix. In addition, the surrounding tissues could affect the absorption of the drug, resulting in a relatively smaller gravity-induced concentration gradient. Furthermore, errors in the measurement of bevacizumab concentration due to inevitable mixing of samples while collecting a relatively small amount of sample (0.2 mL) from each layer should be taken into consideration. As the sample was collected by passing through the upper layer of the solution, we cannot completely rule out a possibility that this method may have affected the lower layer's bevacizumab concentration even if microcapillary tips had been used carefully. However, we believe it is very unlikely that this sampling method, would have affected our results that the lower layer's bevacizumab concertation was statistically significantly higher because the upper layer was found to have a lower bevacizumab concentration. Nevertheless, given that the patient must rest in a fixed position after vitrectomy, we believe that bevacizumab will be present at a higher concentration in the inferior part of the vitreous chamber at least in the early stage.

Further studies on the effect of gravity on bevacizumab, including model eye, animal experiments, and clinical trials, are required in the future. The results of the present *in vitro* distribution study can lay the foundation for future studies. In addition, further research should be conducted to verify whether this gravity-dependent distribution is evident for intraocular bevacizumab injection in patients who have not undergone vitrectomy. This would help determine the optimal postoperative head position for the vast number of patients who undergo intraocular bevacizumab injection worldwide.

## Supporting information

**S1 Table. The details of individual concentration measurements for each tube at each time point.**
(XLSX)

## Author Contributions

**Conceptualization:** Ho Ra.

**Data curation:** Rae Young Kim.

**Formal analysis:** Rae Young Kim.

**Investigation:** Rae Young Kim.

**Methodology:** Rae Young Kim.

**Project administration:** Ho Ra.

**Supervision:** Soonil Kwon, Ho Ra.

**Validation:** Soonil Kwon, Ho Ra.

**Visualization:** Rae Young Kim, Soonil Kwon.

**Writing – original draft:** Rae Young Kim.

**Writing – review & editing:** Soonil Kwon, Ho Ra.

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
