## [Decision Letter · Decision Letter 0]

22 Jul 2019

PONE-D-19-17798

Gravity influences bevacizumab distribution in an undisturbed balanced salt solution in vitro

PLOS ONE

Dear A/porf. Ra,

Thank you for submitting your manuscript to PLOS ONE. After careful consideration, we feel that it has merit but does not fully meet PLOS ONE’s publication criteria as it currently stands. Therefore, we invite you to submit a revised version of the manuscript that addresses the points raised during the review process.

We would appreciate receiving your revised manuscript by Sep 05 2019 11:59PM. To enhance the reproducibility of your results, we recommend that if applicable you deposit your laboratory protocols in protocols.io, where a protocol can be assigned its own identifier (DOI) such that it can be cited independently in the future. For instructions see: http://journals.plos.org/plosone/s/submission-guidelines#loc-laboratory-protocols

We look forward to receiving your revised manuscript.

Kind regards,

M. Elizabeth Hartnett

Academic Editor

PLOS ONE

Journal Requirements:

2. Please include the underlying raw data in a Supporting Information file, namely the individual concentration measurements for each tube at each timepoint.

Additional Editor Comments:

Please address concerns of reviewers 1 and 2 carefully especially the concern about using test tubes and how clinically applicable your study is.

Please also temper your conclusions based on the concerns of the reviewers.

Reviewers' comments:

Reviewer's Responses to Questions

**Comments to the Author**

1. Is the manuscript technically sound, and do the data support the conclusions?

Reviewer #1: Yes

Reviewer #2: No

Reviewer #3: Yes

2. Has the statistical analysis been performed appropriately and rigorously? 

Reviewer #1: Yes

Reviewer #2: Yes

Reviewer #3: I Don't Know

3. Have the authors made all data underlying the findings in their manuscript fully available?

Reviewer #1: Yes

Reviewer #2: Yes

Reviewer #3: Yes

4. Is the manuscript presented in an intelligible fashion and written in standard English?

Reviewer #1: Yes

Reviewer #2: Yes

Reviewer #3: Yes

5. Review Comments to the Author

Reviewer #1: The findings of the authors are interesting. However, based on the data presented, there seems to be no way to apply these purely in vitro findings to clinical practice at this time. Therefore, the concluding sentence of results is an overstatement and not supported by the results presented, “These factors should be taken into account during intraocular bevacizumab injection after vitrectomy.” Recommend this be eliminated or modified substantially to directly and accurately reflect the results presented.

Similarly for the comments made in lines 137-138: there is no data in the current manuscript to support the claim that “similar results would be observed in clinical practice.” This should be removed or substantively modified. Also for lines 153-154, there is no data to indicate that the current results “need to be taken into account during intraocular injection.” Again, over reach by the authors beyond a reflection of their actual, presented data. It would be acceptable to mention these issues as possibilities requiring more study, particularly in vivo studies. But, to conclude that human treatment should be directly modified based on this work alone should be avoided.

The authors state that a “micropipette” was used to collect the samples from the tubes at the end of the prespecified waiting period. How was this performed for the “lower layer” without disturbing the “upper layer” in the process. Or, was all of the upper layer removed and then the lower layer was accessed? Please clarify exactly how the samples were harvested prior to ELISA analysis. This seems to be partially addressed in lines 163-166, but clarity in the Methods would be valuable to readers.

Reviewer #2: The study aims to evaluate the effect of gravity on bevacizumab in vitro.The authors added bevacizumab to the upper part of a test tube filled with balanced salt solution (BSS) and examined its distribution over time. They found significant differences in concentration between the upper and lower parts of the test tubes, even after a considerable amount of time had passed which shows that bevacizumab did not dissolve immediately and diffuse evenly throughout the solution. They reasoned that more bevacizumab settles in the lower part of the tube relative to the upper part because of gravitational force.

The biggest problem of this study is they used test tubes to conduct the study which is so different from the actual environment in the eye. This will make the data difficult to be interpreted in the "in vivo" condition.

Other comments:

1. Please describe the material of test tubes, which will be a possible factor to affect the distribution of medications

2. Please describe the meaning of all the abbreviations

3. In discussion, authors stated that bevacizumab was injected at a constant injection speed, however, in material and methods parts, the way of injection was not described, which would also cause the difference in distribution. Please also state further how to keep a constant injection speed.

4. The method of temperature control should be mentioned

5. In the first paragraph of discussion, authors stated that the concentration was higher in the upper part after 6 to 168 hours after injection, however, the first measurement time stated in your methods was 12 hours after the injection.

Minor points:

6. Line 56: "bevacizumab injection is often prescribed after vitrectomy", may change to "Bevacizumab injection is often given, or performed, after vitrectomy

7. Line 60: “whether bevacizumab dissolves and disperses rapidly and evenly throughout the vitreous chamber when injected into balanced salt solution (BSS) filling the vitreous chamber after vitrectomy”, may change to “when injected into balanced salt solution (BSS) filled eyes”

8. Line 65: “Through this, we aimed…”, you can delete the word “Through this” and start the paragraph with “We aimed…”

9. Line 90: “Bevacizumab concentrations...” should be written as “Bevacizumab’s concentrations...”

Reviewer #3: I believe this research may have other applications besides DR. For instance, Rop and Srn.

Can the authors please comment on thoughts on the same concept with an intact vitreous?

It would be interesting to do an animal model for the distribution of antivegf in eyes with intact vitreous.

6. PLOS authors have the option to publish the peer review history of their article (what does this mean?). If published, this will include your full peer review and any attached files.

Reviewer #1: No

Reviewer #2: No

Reviewer #3: Yes: Clio Armitage Harper III

---

## [Author Response · Author response to Decision Letter 0]

19 Sep 2019

Response to reviewers

Reviewer #1: The findings of the authors are interesting. However, based on the data presented, there seems to be no way to apply these purely in vitro findings to clinical practice at this time. Therefore, the concluding sentence of results is an overstatement and not supported by the results presented, "These factors should be taken into account during intraocular bevacizumab injection after vitrectomy." Recommend this be eliminated or modified substantially to directly and accurately reflect the results presented. We agree with your comment and therefore, we have deleted the relevant sentence from the Abstract. 

Similarly for the comments made in lines 137-138: there is no data in the current manuscript to support the claim that "similar results would be observed in clinical practice." This should be removed or substantively modified.  We have revised the relevant sentence. Also for lines 153-154, there is no data to indicate that the current results "need to be taken into account during intraocular injection."  We have made the necessary revision. Again, over reach by the authors beyond a reflection of their actual, presented data. It would be acceptable to mention these issues as possibilities requiring more study, particularly in vivo studies. But, to conclude that human treatment should be directly modified based on this work alone should be avoided.

The authors state that a "micropipette" was used to collect the samples from the tubes at the end of the pre-specified waiting period. How was this performed for the "lower layer" without disturbing the "upper layer" in the process. Or, was all of the upper layer removed and then the lower layer was accessed? Please clarify exactly how the samples were harvested prior to ELISA analysis. This seems to be partially addressed in lines 163-166, but clarity in the Methods would be valuable to readers. Thank you for your comment. To collect the lower layer, microcapillary tips (Denville Scientific, Metuchen, NJ) were inserted slowly and carefully to ensure that the upper layer was not mixed with the lower layer. We have described this in the Methods.

As we collected the sample from the lower layer by passing through the upper layer of the solution, we cannot completely rule out the possibility that this method may have affected the lower layer’s bevacizumab concentration. However, we believe that it is highly unlikely that this sampling method would have affected our results, as the lower layer’s bevacizumab concertation was statistically significantly higher. We have described this in the Discussion. 

Reviewer #2: The study aims to evaluate the effect of gravity on bevacizumab in vitro. The authors added bevacizumab to the upper part of a test tube filled with balanced salt solution (BSS) and examined its distribution over time. They found significant differences in concentration between the upper and lower parts of the test tubes, even after a considerable amount of time had passed which shows that bevacizumab did not dissolve immediately and diffuse evenly throughout the solution. They reasoned that more bevacizumab settles in the lower part of the tube relative to the upper part because of gravitational force.

The biggest problem of this study is they used test tubes to conduct the study which is so different from the actual environment in the eye. This will make the data difficult to be interpreted in the "in vivo" condition. We have toned down the Conclusions to better reflect the results. We have also stated our limitations and mentioned that animal experiments or in vivo clinical trials are necessary.

Other comments:

1. Please describe the material of test tubes, which will be a possible factor to affect the distribution of medications  We have added the necessary information in the second line of the Methods. 

2. Please describe the meaning of all the abbreviations

3. In discussion, authors stated that bevacizumab was injected at a constant injection speed, however, in material and methods parts, the way of injection was not described, which would also cause the difference in distribution. Please also state further how to keep a constant injection speed. Thank you for your detailed review. We have provided the necessary information in the Methods.

4. The method of temperature control should be mentioned-> We have provided the necessary information in the Methods.

5. In the first paragraph of discussion, authors stated that the concentration was higher in the upper part after 6 to 168 hours after injection, however, the first measurement time stated in your methods was 12 hours after the injection.  Originally, we collected data from 6 h, but we just indicated data from 12 h because it did not affect our results. Thank you for pointing out our mistake; we have made the necessary change.

Minor points:

6. Line 56: "bevacizumab injection is often prescribed after vitrectomy", may change to "Bevacizumab injection is often given, or performed, after vitrectomy -> We have made the necessary change. 

7. Line 60: "whether bevacizumab dissolves and disperses rapidly and evenly throughout the vitreous chamber when injected into balanced salt solution (BSS) filling the vitreous chamber after vitrectomy", may change to "when injected into balanced salt solution (BSS) filled eyes" -> We have made the necessary change.

8. Line 65: "Through this, we aimed…", you can delete the word "Through this" and start the paragraph with "We aimed…" -> We have made the necessary change.

9. Line 90: "Bevacizumab concentrations..." should be written as "Bevacizumab's concentrations..." -> We have revised the relevant sentence.

Reviewer #3: I believe this research may have other applications besides DR. For instance, Rop and Srn.

Can the authors please comment on thoughts on the same concept with an intact vitreous?

It would be interesting to do an animal model for the distribution of antivegf in eyes with intact vitreous.

 -> Thank you for your detailed review.

We are planning to conduct additional animal experiments or in vivo clinical trials.

This study's experimental conditions are not identical to real-world clinical situations, but the results of the present in vitro distribution study lay a foundation for future studies.

---

## [Editor Report · Decision Letter 1]

23 Sep 2019

Gravity influences bevacizumab distribution in an undisturbed balanced salt solution in vitro

PONE-D-19-17798R1

Dear Dr. Ra,

We are pleased to inform you that your manuscript has been judged scientifically suitable for publication and will be formally accepted for publication once it complies with all outstanding technical requirements.

With kind regards,

M. Elizabeth Hartnett

Academic Editor

PLOS ONE
---

## [Editor Report · Acceptance letter]

27 Sep 2019

PONE-D-19-17798R1 

Gravity influences bevacizumab distribution in an undisturbed balanced salt solution *in vitro*

Dear Dr. Ra:

I am pleased to inform you that your manuscript has been deemed suitable for publication in PLOS ONE. Congratulations! Your manuscript is now with our production department. 

With kind regards,

on behalf of

Dr. M. Elizabeth Hartnett 

Academic Editor

PLOS ONE